# Risk-Benefit Assessment of Cereal-Based Foods Consumed by Portuguese Children Aged 6 to 36 Months—A Case Study under the RiskBenefit4EU Project

**DOI:** 10.3390/nu13093127

**Published:** 2021-09-08

**Authors:** Ricardo Assunção, Géraldine Boué, Paula Alvito, Roberto Brazão, Paulo Carmona, Catarina Carvalho, Daniela Correia, Paulo Fernandes, Carla Lopes, Carla Martins, Jeanne-Marie Membré, Sarogini Monteiro, Pedro Nabais, Sofie T. Thomsen, Duarte Torres, Sara M. Pires, Lea S. Jakobsen

**Affiliations:** 1Food and Nutrition Department, National Institute of Health Dr. Ricardo Jorge, 1649-016 Lisboa, Portugal; paula.alvito@insa.min-saude.pt (P.A.); roberto.brazao@insa.min-saude.pt (R.B.); carla.martins@ensp.unl.pt (C.M.); 2CESAM, Centre for Environmental and Marine Studies, University of Aveiro, 3810-193 Aveiro, Portugal; 3IUEM, Instituto Universitário Egas Moniz, Egas Moniz-Cooperativa de Ensino Superior, CRL, 2829-511 Caparica, Portugal; 4INRAe, Oniris, Secalim, 44307 Nantes, France; geraldine.boue@oniris-nantes.fr (G.B.); paulo.fernandes@insa.min-saude.pt (P.F.); jeanne-marie.membre@oniris-nantes.fr (J.-M.M.); 5Food Risks Unit, Economic and Food Safety Authority (ASAE), 1649-038 Lisboa, Portugal; pjcarmona@asae.pt (P.C.); scmonteiro@asae.pt (S.M.); pmnabais@asae.pt (P.N.); 6Faculty of Nutrition and Food Sciences, University of Porto, 4150-180 Porto, Portugal; catarinacarvalho@fcna.up.pt (C.C.); dupamato@fcna.up.pt (D.T.); 7EPIUnit—Institute of Public Health, University of Porto, 4050-600 Porto, Portugal; danielamc@med.up.pt (D.C.); carlal@med.up.pt (C.L.); 8Department of Public Health and Forensic Sciences and Medical Education, Faculty of Medicine, University of Porto, 4200-450 Porto, Portugal; 9NOVA National School of Public Health, Public Health Research Center, Universidade NOVA de Lisboa, 1600-560 Lisboa, Portugal; 10Division for Diet, Disease Prevention and Toxicology, The National Food Institute, Technical University of Denmark, 2800 Kgs Lyngby, Denmark; sthth@food.dtu.dk (S.T.T.); smpi@food.dtu.dk (S.M.P.); leaja@food.dtu.dk (L.S.J.)

**Keywords:** public health, risk–benefit assessment, cereal-based foods, children, mycotoxins, *Bacillus cereus*, sodium, fiber, free sugars

## Abstract

Cereal-based foods, including breakfast (BC) and infant cereals (IC), are among the first solid foods introduced to infants. BC and IC are sources of macro and micronutrients that have beneficial effects on health, but can also be sources of harmful chemical and microbiological contaminants and nutrients that may lead to adverse health effects at high consumption levels. This study was performed under the RiskBenefit4EU project with the aim of assessing the health impact associated with consumption of BC and IC by Portuguese children under 35 months. Adverse effects associated with the presence of aflatoxins, *Bacillus cereus*, sodium and free sugars were assessed against the benefits of fiber intake. We applied a risk–benefit assessment approach, and quantified the health impact of changes in consumption of BC and IC from current to various alternative consumption scenarios. Health impact was assessed in terms of disability-adjusted life years. Results showed that moving from the current consumption scenario to considered alternative scenarios results in a gain of healthy life years. Portuguese children can benefit from exchanging intake of IC to BC, if the BC consumed has an adequate nutritional profile in terms of fiber, sodium and free sugars, with levels of aflatoxins reduced as much as possible.

## 1. Introduction

Cereal-based foods, including breakfast cereals (BC), i.e., ready-to-eat processed grains; and infant cereals (IC), i.e., processed cereal-based foods that are or have to be reconstituted with water, milk, or other appropriate liquids, are among the first solid foods introduced to infants (age 0–1 (www.cdc.gov (accessed on 19 August 2021))) and are important components of the diet for toddlers (age 0–3 (www.cdc.gov (accessed on 19 August 2021))) [1,2]. They are sources of nutrients necessary for a healthy development, including macronutrients (e.g., fiber) and essential micronutrients (e.g., vitamin B2, potassium, magnesium). Cereal-based foods are also an important source of whole grains [3,4]. Additionally, some BCs and ICs may be fortified with vitamins such as folate and thiamine. However, cereal-based foods can also be a vehicle for hazardous contaminants or other components that can potentially lead to adverse health effects. Cereals can be contaminated with harmful chemicals such as mycotoxins [5,6,7,8], heavy metals [9,10,11,12,13], acrylamide [14], furans [15] and polycyclic aromatic hydrocarbons (PAHs) [16] that are associated with acute and chronic toxic effects, such as gastroenteritis, growth impairment, immune-toxicity and cancer. Despite being considered low moisture foods (LMF), food monitoring has shown that cereals may be contaminated with *Bacillus cereus*, a Gram-positive, rod-shaped, spore-forming bacterium. *B. cereus* can lead to two types of illness in humans: emetic (caused by pre-formed toxins in food) and diarrheal (caused by ingestion of a large number of bacterial cells), and foodborne outbreaks have identified cereals as a vehicle [17]. Depending on their recipes, cereal-based foods may also be a key source of sodium and free sugars, which, at high consumption levels, increase the risk of type 2 diabetes, cardiovascular disease and cancer [18,19].

In Portugal, previous studies have reported that ICs and BCs constitute a considerable proportion of the diet of infants and toddlers (children under three years of age) [5,6,20,21,22]. ICs are traditionally considered suitable as the first solid foods introduced to infants at 6 months of age. BCs are also marketed for infants but considered more suitable for toddlers of older age. According to products available on the Portuguese market, BCs usually contain several types of grains, more fiber and less sodium and added sugar than ICs [5,6,22]. Previous studies have demonstrated the occurrence of mycotoxins in both BCs and ICs available on the Portuguese market [5,6,7,23,24] and the risk associated with exposure to aflatoxins through consumption of cereal-based foods [6]. Likewise, the consumption of IC and BC by young children could represent a significant contribution to the daily intake of sodium and sugar, but also to overall energy intake (recommended energy intake for children up to 36 months is, according to Food and Agriculture Organization (FAO), up to 427 KJ/kg bodyweight per day [25]), and act as risk factors for the development of future non-communicable diseases. Therefore, it is not straightforward to predict the health impact following a change in consumption of BC and IC, and an approach allowing for the integration of associated health risks and benefits will be needed in order to inform dietary recommendations that maximize health benefits for children. 

Risk–benefit assessment (RBA) of foods aims to assess the combined adverse and beneficial health effects associated with foods. RBA integrates chemical, microbiological and nutritional risk and benefit assessments [26,27,28,29] and is used to inform food safety and public health strategies, including updating dietary advice [30]. “RiskBenefit for EU—Partnering to strengthen the risk–benefit assessment within EU using a holistic approach” (RB4EU) was a knowledge translation project funded by the European Food Safety Authority (EFSA), which gathered a multidisciplinary team and developed a framework for capacity building in RBA through case studies [26,31].

The aim of this case study was to perform an RBA to quantify the overall health impact in terms of DALYs from changes in the consumption of BC and IC by Portuguese children below 36 months of age. Disability-Adjusted Life Years (DALYs) are a composite health metric commonly applied to integrate risks and benefits in an estimate of overall health gain or loss following dietary changes [32]. We followed the harmonized framework developed under the RiskBenefit4EU project to answer the question: how many healthy life years are gained or lost by changes in the consumption of BC and IC by Portuguese children below 35 months of age?

## 2. Materials and Methods

### 2.1. Selection of Relevant Components and Health Effects

A two-step literature search was carried out to identify the BC and IC components (contaminants/nutrients) and associated health effects to be considered in the study: step (1) a “food–component” literature search, to identify components of interest for the food products considered; step (2) a “component-health effect” literature search, to identify the health effects associated with specific food components. The search strategies, identified components and health effects are summarized in Appendix B. Table 1 summarizes the selected food components, their associated health effects and the type of analysis performed. 

Mycotoxins are widely occurring secondary metabolites produced by fungi that can contaminate foods and feeds. Aflatoxins, a subgroup of mycotoxins, are genotoxic, carcinogenic and immunosuppressive substances. The four major aflatoxins are B_1_, B_2_, G_1_ and G_2_. Aflatoxin B_1_ (AFB_1_) is the most potent naturally occurring chemical liver carcinogen known. These toxins cause hepatocellular carcinoma (HCC) [33] and thus have been classified as Group 1 carcinogens (carcinogenic to humans) by the International Agency for Research on Cancer [34]. HCC is the third leading cause of cancer deaths worldwide and was estimated to have been responsible for nearly 746,000 deaths in 2012 (9.1% of all cancer deaths that year) [35]. 

*B. cereus* is a foodborne pathogen that causes diarrhea and vomiting [36], and is the most frequent causative agent of foodborne illnesses in cereals and grains. It was responsible for foodborne outbreaks associated with exposure to cereal products in different regions of the United States, Australia, New Zealand, Asia and Europe [17]. Some of the reported outbreaks involved IC [36] and ready-to-eat cereals [37,38]. 

Dietary fiber may contribute to the prevention of childhood obesity, maintenance of normal blood glucose, lipids and blood pressure, however, the scientific evidence is rather contradictory for some of these outcomes [39]. On the other hand, there is convincing evidence that dietary fiber intake promotes normal gastrointestinal function, especially laxation [40], and convincing/probable evidence that it reduces the risk of colon cancer [41,42,43]. Moreover, probable evidence shows that fiber intake reduces the risk of cardiovascular disease (CVD) and type 2 diabetes (DM2) [40,41,43,44]. Children with higher intakes of dietary fiber tend to consume diets that are more rich in nutrients and are more likely to meet recommended daily intakes for key nutrients [45]. Cereal-based foods with a high content of wholegrains constitute an important source of dietary fiber. 

Sodium is an essential cation that is necessary for normal cell function and for neurotransmission. However, high sodium intake is associated with increased blood pressure among adults and children and with a high risk of cardiovascular diseases among adults. The World Health Organization (WHO) recommends a maximum sodium intake of 2 g/day in adults. For children, this value should be adjusted downward based on the energy requirements of children when compared to adults [18]. Accordingly, EFSA recommends an intake of 1.1 g of sodium per day as safe and adequate for children aged 1–3 years old [46].

The WHO defines free sugars as all monosaccharides and disaccharides added to foods (i.e., added sugars), and sugars naturally present in honey, syrups, and fruit juices. Moderate-quality evidence suggests that reduced intake of free sugars is associated with reduced body weight in children [19]. However, higher rates of dental caries are observed when free sugar intake exceeds 10% of total energy intake [19]. Thus, the WHO recommends that free sugars should not exceed 10% of total energy intake, with the conditional recommendation to reduce free sugar intake to below 5% of total energy intake [19].

Finally and according to Table 1, total aflatoxin-HCC, *B. cereus*-gastrointestinal disease and dietary fiber-DM2 & CVD were included in the quantitative RBA along with sodium and free sugars, in a semi-quantitative approach comparing consumption scenarios using dietary reference values (DRV).

### 2.2. Investigated Consumption Scenarios

In order to assess health impact due to changes in BC and IC consumption, we investigated four scenarios: 100% BC scenario, simulating that all infants consume only BC;100% IC scenario, simulating that all infants consume only IC;Optimal BC scenario, simulating that all infants consume only BC at an optimal composition; andWorst IC, simulating that all infants consume only IC with the worst composition.

Each of the above scenarios was analyzed relative to the current amount consumed, in which only IC, BC or both (i.e., the reference scenario) are consumed by children aged 6 to 35 months, as reported by the National Food, Nutrition and Physical Activity Survey of the Portuguese General Population 2015–2016 (IAN-AF 2015–2016) [22]. Current consumption is defined as the consumption of IC and BC reported in the latest survey of Portuguese dietary habits [22], as described below under 2.3. Specifically, the consumption of IC and BC in scenario 1 and 2 was modelled, by substituting the current consumption of IC by BC and BC by IC, respectively, based on the observed distribution of consumed IC and BC [22]. For scenarios 3 and 4 the best BC and worst IC products were identified among the BC and IC products available on the market, by scoring each product according to their content of fiber, sodium and free sugar per 100 kcal (Appendix B). To create the score, the variables fiber, sodium and free sugar per 100 kcal were first standardized and then summed. Fiber was considered a positive parameter; sodium and free sugar negative ones. The score was calculated for each product as follows:(1)$$Score=z_{fiber}-(z_{sodium}+z_{free\;sugar})$$

The best BC was the one with the highest score among all BC products and the worst IC was the one with the lowest score among all IC products (Table 2). Scenarios 3 and 4 were modelled by substituting the current consumption of IC and BC by the BC with the highest score (“Best BC”) and by the IC with the lowest score (“Worst IC”), respectively. Substitutions in all four scenarios were iso-caloric. 

### 2.3. Data Used in the Model

Food consumption data were collected from the National Food and Physical Activity Consumption Survey (IAN-AF 2015–2016), a representative sample of the Portuguese general population aged between 3 months and 84 years old, with a total of 5811 participants [20,21,22]. In this study, food consumption data from 779 children aged 6 to 36 months from the IAN-AF 2015–2016 sample was included. The sample size of participants was determined, considered in the power analysis, and presented in detail in the methodological publication of the IAN-AF 2015–2016 survey [21]. Data on food consumption were collected according to European guidelines [47]. For children aged under 10 years old, dietary intake was assessed by two non-consecutive one-day food diaries that were filled in by the main caregiver, followed by a face-to-face interview conducted with the caregivers to collect additional details on food description and quantification. The survey used an extended version of the Portuguese Food Composition Table [48] to convert the reported food items into nutrients, covering 38 BCs and 87 ICs.

Concentration data of aflatoxins in ICs and BCs available on the Portuguese market were collected from published surveys [5,6,7]. Detailed information about analytical conditions and method performance of aflatoxin determination were described elsewhere [6,7]. Occurrence levels of aflatoxins in both BCs and ICs were below legislative limits (infant cereals: 0.100 μg/kg for AFB_1_; BC: 2.0 μg/kg for AFB_1_ and 4.0 μg/kg for the sum of AFB_1_, AFB_2_, AFG_1_ and AFG_2_) [49]. 

Prevalence and level of contamination of *B. cereus* in ICs and BCs were modeled using results from the Portuguese National Sampling Plan (PNCA) for BC [50] and available data from the scientific literature for IC [37]. Between 2018 and 2019, the PNCA analyzed 50 samples of BC, corresponding to 30 different batches, for which 5 boxes were tested. Fourteen out of 150 samples had positive levels of *B. cereus*. For IC, data from another study carried out in Portugal in 2007 were applied, in which 35 samples were analyzed, including 15 positives [37]. For both IC and BC, samples below the limit of detection (LOD) (<10 colony forming units (CFU)/g), were assumed to have 5 CFU/g. For BC, positive samples below 40 CFU/g were reported as <40 CFU/g in the database; these were assumed to have 25 CFU/g, i.e., the median between 10 and 40. 

Regarding DALY, this metric combines information on disease incidence, severity, duration and mortality in one number and enables comparison across diseases [51,52]. The epidemiological and toxicological data used to derive the correspondent DALY are presented in Table 3. 

### 2.4. Exposure Assessment, DALY Calculation and Integration of Risks and Benefits 

We applied different approaches for exposure and health impact assessment for each component–health effect pair. 

All models were developed and implemented in the @Risk^®^ software for Microsoft Excel version 6 (Palisade Corporation, Ithaca, NY, USA). We defined model variables as probabilistic distributions to quantify uncertainty. The approach applied for each component is described below and all models are documented in the Appendix A. SPADE software [55] was used to estimate the intake distributions of total fiber, sodium and free sugars in the current and each alternative scenario. 

#### 2.4.1. Total Fiber, Sodium and Free Sugars 

In order to estimate the relative risk (RR) of the current and each alternative intake scenario, we performed the following calculations. The distribution of fiber intake of children aged 6 to 35 months, which was derived from the National Food and Physical Activity Consumption Survey [22], was divided into quartiles, with each quartile representing a consumption class (1–4). The median of each class represented the intake of each class, respectively. RRs for DM2 and CVD derived from the literature (Table 3) were used to estimate an RR for each class, assuming an RR of 1 at zero exposure and a log-linear association between exposure and RR [56] (Appendix C and Appendix A). Thus, the log-linear slope, β, and RR for each class, j ∈ {1, 2, 3, 4}, in each scenario, i ∈ {1, 2, 3, 4, 5} (i.e., the reference and the four alternative scenarios), were calculated according to the following equations:(2)$$\beta =\frac{\ln RR_{literature}}{Dose}$$
(3)$$RR_{i}=\exp\left(\beta \cdot exposure_{i}\right)$$
where *RR_literature_* is the RR for DM2 and CVD per *Dose* reported in the literature (Table 3), and *RR_i_* and *exposure_i_* are the RR and intake of fiber in each scenario.

The potential impact fraction (PIF) is a measure of the proportional change in disease burden after a change in exposure to a related risk factor—in this case changes in exposure to fiber from the current to alternative scenarios—and was calculated for each alternative scenario [57]. We applied RR shift methodology [57], which assumes that the change in exposure is described by a change in the RR of the categories, while keeping the proportion in each category constant: (4)$$PIF=\frac{\sum_{j=1}^{4}RR_{alt}-\sum_{j=1}^{4}RR_{ref}}{\sum_{j=1}^{4}RR_{ref}}$$
where *RR_ref_* is the RR of the reference scenario and *RR_alt_* is the RR of each alternative scenario.

We calculated the incidence and DALYs attributed to the change in IC and BC consumption in each scenario by multiplying incidence and DALY rates for DM2 and CVD for Portugal, obtained from the Global Burden of Disease 2017 study (GBD) [53] (Table 3), by the PIFs for each alternative scenario.

A semi-quantitative analysis was performed for sodium and free sugars. The DRVs used were an upper-limit (UL) of 1500 mg/day for sodium [58] and a recommended intake (RI) of 5% and 10% of total energy intake (TEI) for free sugars [19]. The prevalence of individuals above or below the DRVs was estimated for each scenario using SPADE software [55].

#### 2.4.2. Aflatoxins

We estimated the exposure to aflatoxins from BC and IC consumption by: (5)$$\mathrm{AFT}\;\mathrm{intake}\;(µg/\mathrm{kg}\;\mathrm{bw}/\mathrm{day})=\frac{Conc_{AFT}\times Consump}{bw},\;$$
where *Conc_AFT_* is the concentration of aflatoxins determined in BC and IC in μg/kg; *Consump* is the consumption of BC and IC in kg/day; and bw is the body weight of the considered population in kg.

We used the total concentration of aflatoxins measured in cereal samples, assuming that all aflatoxins have equal carcinogenic potency—a conservative approach previously applied by EFSA [59]. Values below the LOD can be replaced via different approaches as recommended by EFSA: lower-bound (LB, <LOD = 0), middle-bound (MB, <LOD = ½ LOD) and upper-bound (UB, <LOD = LOD) approaches [60]. The different approaches were applied for aflatoxin concentrations in scenarios as follows: (i) an MB approach was used for the reference, 100% IC and 100% BC scenarios; (ii) a LB approach was used for the Best BC scenario, since the best breakfast cereals would have the minimum possible contamination by aflatoxins; (iii) a UB approach was used for the Worst IC scenario, since the worst infant cereal would have the maximum possible contamination by aflatoxins.

We estimated the number of extra HCC cases caused by exposure to aflatoxins from BC and IC in the reference scenario and each alternative scenario by combining the DRV related to aflatoxin exposure and the risk for HCC (Table 3) [54]. The DALYs attributed to the estimated aflatoxin exposure were calculated by multiplying the estimated extra HCC cases by a DALY/case rate, derived from WHO incidence and the DALY envelope [53].

#### 2.4.3. Bacillus Cereus

A beta distribution was used to approximate prevalence based on the percentage of positive values; it was expressed as a beta of “number of positive + 1; total sample—number of positive +1”. For positive samples, to represent the level of *B. cereus*, a cumulative distribution was built for BC and a triangular one for IC. Simulations were run to represent the level of *B. cereus* in IC and BC, giving a value of 5 CFU/g when considered to be non-contaminated (<10 ufc/g) and selecting randomly for positive sample levels from the cumulative or uniform distribution of BC and IC, respectively.

To estimate exposure to *B. cereus* (in CFU/day) through consumption of BC and IC in the considered population group, contamination data in CFU/g and consumption data in g/day were multiplied. This was done by multiple simulations to combine the different possible exposure values (observed for the reference scenario or estimated for alternative scenarios). All variables were defined as probability distributions, informed by the available data. 

Quantitative DR for *B. cereus* were not available in the literature. Consequently, a threshold approach was used considering a potential case of illness when the product contamination exceeded a threshold of 1.0 × 10^3^ CFU/g [61] or when the overall level of daily exposure to bacteria exceeded a threshold of 5.0 × 10^4^ CFU/day [36]. 

#### 2.4.4. Integration of DALY

The difference in DALYs over each health outcome attributed to a change in IC and BC consumption was summed to estimate the overall health impact (∆DALY) of each scenario. ∆DALY > 0 implies a health loss due to the change in consumption. ∆DALY < 0 implies a health gain due to the change in consumption.

## 3. Results

### 3.1. Intake and Exposure Assessment

The estimated median daily consumption of BC and IC by Portuguese children between 6 and 35 months of age (reference scenario) was 0.0 g and 8.7 g, respectively (Table 4). The substitution of the current consumption of IC and BC by the Best BC scenario resulted in the highest daily consumption (14.5 g/day) among the alternative scenarios. 

The Best BC scenario resulted in a mean fiber intake of 11.3 g/day, i.e., around 2 g/day higher than the other scenarios (Table 5). Regarding sodium and free sugars, intake was estimated to be lower in the Best BC scenario when compared to the remaining scenarios (Table 5). The Best BC scenario also resulted in the lowest mean exposure to aflatoxins. This scenario also reflects a full reduction in exposure to *B. cereus*, as no contaminated samples were found where it could be targeted, contrary to the Worst IC scenario, which led to around 3.3 log CFU/day (Table 5).

### 3.2. Incidence and DALYs

For DM2 and CVD, the Best BC scenario presented the highest number of cases prevented by change in fiber intake from the reference scenario. The Worst IC scenario presented the highest number of extra cases caused by change in fiber intake (Table 6). The highest number of extra cases of HCC was estimated for the Worst IC scenario (Table 6). Regarding *B. cereus*, no cases of gastrointestinal illnesses were estimated as all scenarios provided exposure levels far below both thresholds of illness; all simulations estimated exposure levels below the threshold of 5.0 × 10^4^ CFU/day [36] and product concentrations below the maximum limit of 1.0 × 10^3^ CFU/g [61].

Figure 1 presents the ΔDALY across the health outcomes accounted for in each domain, i.e., nutrition (fiber), toxicology (aflatoxins) and microbiology (*B. cereus)* for the different scenarios assessed. As no cases of gastroenteritis due to *B. cereus* were identified in any of the scenarios, no DALYs were likewise estimated. The 100% BC and Best BC scenarios resulted in negative ΔDALY for fiber, indicating a health gain associated with a change in intake to these scenarios from the current scenario. On another hand, the 100% IC and Worst IC scenarios resulted in a positive ΔDALY, indicating a health loss due to the change in intake (Figure 1). Contribution from aflatoxins was negligible, accounting for a ΔDALY close to zero in all the considered scenarios.

When integrating ΔDALY over the domains, an overall ΔDALY of 0.143 and 0.877 per 100,000 individuals in the 100% IC scenario and Worst IC scenario, respectively, was estimated. In contrast, the 100% BC and Best BC scenarios resulted in an overall negative ΔDALY of −0.486 and −4.473, respectively (Table 7).

### 3.3. Comparison to DRVs

Table 8 presents the prevalence of inadequate intake of sodium and free sugars when compared to DRVs. These results reflect the intake of sodium and free sugars from all dietary sources, and not only the contribution of IC and BC in all the scenarios considered. We estimated that 25% of Portuguese children between 6 and 35 months of age exceed the upper-limit (UL) intake for sodium in the reference scenario, where BC and IC contribute 0.7% and 1.3% to total sodium, respectively. This prevalence was increased slightly in the 100% BC and Worst IC scenarios compared to the reference scenario (from 24.8% to 25.7% and 25.4%, respectively). In the Best BC scenario, the prevalence was reduced to 24% over the UL. Currently, 29% have an intake of free sugars higher than 10% of TEI. The prevalence of individuals exceeding the RI was lowest (16%) in the Best BC scenario. For all scenarios, the prevalence of exceeding the RI of free sugars considerably increased if 5% TEI was considered as the cut-off. 

## 4. Discussion

### 4.1. Consumption of Breakfast Cereals Instead of Infant Cereals Could Result in a Gain of Healthy-Life Years

This study assessed the risk–benefit balance of consuming different proportions of BC and IC by Portuguese children aged between 6 and 35 months. The health risks associated with the presence of aflatoxin and *B. cereus* were evaluated as well as the benefits of the intake of fiber, by comparing four different scenarios of BC and IC consumption with current consumption. Overall health impact was expressed in DALYs. The change in the intake of sodium and free sugars from current to alternative consumption scenarios were evaluated by a semi-quantitative approach. We found that the substitution of current consumption by 100% BC and Best BC would lead to a gain in healthy-life years. The Best BC scenario improved the percentage of the considered population exceeding the UL and RI for sodium and free sugars, respectively. To our knowledge, this is the first RBA of cereal-based foods intended for children aged between 6 and 35 months.

Our results suggest that children can benefit from exclusive BC consumption if product formulations are optimized in terms of fiber, sodium and free sugar content, and at the same time, aflatoxin levels are reduced as much as possible. Choosing the best BC available in the Portuguese market would benefit population health in terms of prevention of DM2 and CVD and a decrease in the prevalence of exceeding the UL and RI of sodium and free sugars, respectively. The consumption of this cereal-based product could prevent a loss of about 4.5 healthy life years per 100,000 individuals annually. However, BCs with high fiber and low sodium and free sugars content could lead to low palatability and consequently a low adherence by consumers, especially for the considered age group. Our results also showed that the negative impact on health due to exposure to aflatoxins was minor in all scenarios considered. However, previous studies performed in Portugal [5,6] and abroad [62] have stressed the risk associated with the consumption of cereal-based foods, including BC and IC, due to exposure to aflatoxins present in these food products—showing, for the highest percentiles of intake, a margin of exposure below 10,000, representing a potential health risk [5,6]. Cereal-based products constitute one of the main contributors of human exposure to mycotoxins and the Food and Agriculture Organization of the United Nations (FAO) estimated that at least 25% of cereals produced in the world are contaminated by mycotoxins, acknowledging that this value could be higher than 50% if emerging mycotoxins are considered, but of which limited data are so far available [63,64]. Mycotoxins are natural contaminants and cannot be completely prevented. Consequently, significant efforts should be considered, from farm to fork, to improve the safety and quality of foods, reducing mycotoxin exposure—particularly to carcinogenic aflatoxins. 

Recognizing the potential benefits, there is a growing interest in the reformulation of breakfast cereal products [65]. European countries have been developing and establishing programs to reduce the intake of sodium and sugar. In Italy, in an attempt to improve the nutritional characteristics of food products, one objective was to reformulate breakfast cereals by reducing sugar and salt down to mean contents of 30 and 0.4 g per 100 g, respectively, and increasing fiber to 4.5 g per 100 g [66]. In Portugal, the Integrated Strategy for the Promotion of Healthy Eating was adopted in 2017, which also considered promotion of the reformulation of some food categories, including breakfast cereals, by monitoring the salt and sugar in breakfast cereals [67]. In the effort to reformulate BCs, attention should also be paid to aflatoxin levels in these food products. Despite being consumed by this age group in Portugal, breakfast cereals are not recognized as infant foods in a legislative context [49]. If BC products were included in the category “infant foods”, stricter limits would be applicable, and consequently extra efforts to produce cereal-based foods presenting lower contamination levels of mycotoxins would be developed. Other studies based on qualitative assessments argued that the risk of exposure to aflatoxins through cereal-based product consumption is negligible compared to benefits related to nutritional compounds [68,69]. The present study confirms this perspective in a quantitative assessment. However, the ALARA (as low as reasonably achievable) principle needs to be considered to ensure that levels of aflatoxins in foods are kept as low as possible.

Strategies related to the future reformulation of food products should preferably be evaluated in quantitative assessments of the reformulations’ impact on public health. In this study, we included the impact of changes in fiber and aflatoxin exposure in a full quantitative assessment. Due to a lack of data, the health impact associated with intake of free sugars and sodium was assessed semi-quantitatively, not included in the overall DALY estimate. Nevertheless, the potential health effects associated with these two food components are likely important. 

### 4.2. Sources of Uncertainty

In the present study, the analysis of the obtained results should be considered with regard to all sources of uncertainties that were identified. These sources of uncertainty could influence the precise estimate of the health impact of the considered scenarios. Identified sources of unquantified uncertainty are listed below: **Selection of relevant food components and associated health effects**: Despite an extensive literature search being performed to support decisions in the selection of food components and associated health effects, other beneficial (e.g., micronutrients used for fortification or saturated fats) or hazardous (e.g., other mycotoxins such as deoxynivalenol and zearalenone, heavy metals, furans, acrylamide) components could be present in BC and IC; also, food allergies are relevant health outcomes when investigating the health impact of infant and toddler foods, but have been disregarded in this assessment.**Data used in the exposure assessment**: Concentration of *B. cereus* and aflatoxins in BC and IC were based on reported levels of analyzed samples or extracted from the Portuguese Food Composition Table (sodium, free sugars and fiber). These data are necessarily affected by uncertainty that impacted (over and/or underestimated) the exposure assessment.**Risk of CVD and T2D later in life based on exposure that occurred in the first years of life**: Due to the restrictions of the data, CVD and T2D cases were estimated, taking into account the available dose–response data. However, it was assumed that exposure occurring in the first years of life will contribute to and determine cases later in life.

## 5. Conclusions

This study demonstrated that changes in consumption patterns of BC and IC in children below 35 months of age in Portugal would lead to health gains in the population. These results can be considered an alert for industry and risk managers, motivating strategies and measures for reduction of aflatoxins in breakfast cereals, and supporting higher content of fiber. Efforts must also continue to maintain low levels of *B. cereus* contamination, and to further reduce sodium and free sugar content in recipes. A combination of these actions will contribute to improve children’s health during childhood as well as in the long-term. Application of the RB4EU framework for the risk–benefit assessment proved useful, and we encourage its use to address other risk–benefit questions.

## Figures and Tables

**Figure 1 nutrients-13-03127-f001:**
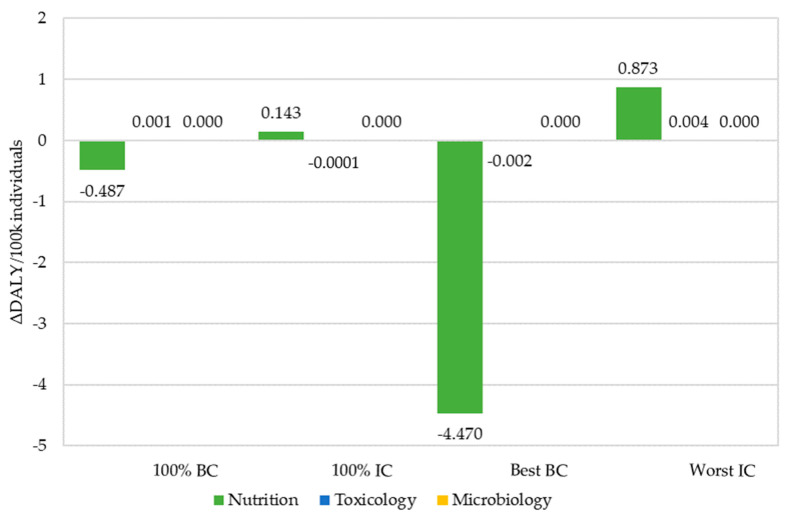
Health impact per domain of nutrition, toxicology and microbiology in each alternative scenario expressed in DALYs/100,000/year. BC: breakfast cereals; IC: infant cereals.

**Table 1 nutrients-13-03127-t001:** Food components and health effects considered for the risk–benefit assessment.

	Food Components	Health Effects	Type of Analysis
Nutrition	Total fiber	Type 2 Diabetes mellitus Cardiovascular diseases	Quantitative ^a^
	Sodium	Type 2 diabetes	Semi-quantitative ^b^
	Free sugars	Cardiovascular disease Cancer (different organs)	
Toxicology	Aflatoxins (AFB_1_) ^c^	Hepatocellular carcinoma	Quantitative ^a^
Microbiology	*Bacillus cereus*	Gastrointestinal disease	Quantitative ^a^

^a^ Risks and benefits quantified in DALY; ^b^ Intake compared with the dietary reference values; ^c^ AFB_1_: aflatoxin B_1_.

**Table 2 nutrients-13-03127-t002:** Nutritional composition of “Best Breakfast cereal (BC)” and “Worst Infant cereal (IC)” products selected among the cereal products available on the Portuguese market.

	Fiber (g/100 kcal)	Sodium (mg/100 kcal)	Free Sugar (g/100 kcal)
Best BC	3.9	1.4	0
Worst IC	0.2	51.4	4.2

**Table 3 nutrients-13-03127-t003:** Epidemiological and toxicological data used for the model.

	Dose–Response (Cases/100,000/year/ng/kg AFT bw/day)	Incidence ^2^ (Cases/100 k) (Mean, 95% CI)	Risk Ratios (Mean, 95% CI)	DALY Rate ^2^ (DALY/100 k) (Mean, 95% CI)
Hepatocellular carcinoma (HCC)	0.01 ^3^	NA	NA	9.96
				(7.77–12.61)
Cardiovascular disease (CVD)	NA	25.07	0.78 ^1^	60.78
		(21.36–29.12)	(0.68–0.90)	(49.67–73.44)
Diabetes mellitus 2 (DM2)	NA	168.07	0.85 ^1^	22.4
		(125.70–215.47)	(0.82–0.89)	(12.71–34.47)

NA = Not applicable; CI = Confidence interval; DALY = Disability-adjusted life years; AFT = Aflatoxins; ^1^ per 8 g/day of fiber [39]; ^2^ Global Burden of Disease 2017 study (GBD) [53]; ^3^ WHO (2017) [54].

**Table 4 nutrients-13-03127-t004:** Distribution of consumption represented by the median (P5-P95) of BC and IC consumption by Portuguese children between 6 and 35 months of age, in reference and alternative scenarios.

	Reference	100% BC	100% IC	Best BC	Worst IC
BC (g/day)	0.0 (0.0–20.8)	13.4 (0.0–51.1)	-	14.5 (0.0–55.5)	-
IC (g/day)	8.7 (0.0–42.7)	-	12.5 (0.0–50.1)	-	13.7 (0.0–52.3)

P5: 5th percentile; P95: 95th percentile; BC: breakfast cereals; IC: infant cereals.

**Table 5 nutrients-13-03127-t005:** Usual intake distribution of fiber (g/day), sodium (g/day), free sugars (g/day) and exposure to aflatoxins (ng/kg/ day) and Bacillus cereus (log CFU/day) in Portuguese children between 6 and 35 months of age, in reference and alternative scenarios.

		Reference	100% BC	100% IC	Best BC	Worst IC
Fiber (g/day)	Mean	9.3	9.5	9.2	11.3	8.9
	Median	9.0	9.2	9.0	11.0	8.7
	(P25–P75)	(7.2–11.1)	(7.3–11.4)	(7.1–11.0)	(8.8–13.5)	(6.8–10.7)
Sodium (g/day)	Mean	1.17	1.18	1.16	1.15	1.18
	Median	1.15	1.17	1.14	1.13	1.16
	(P25–P75)	(0.79–1.50)	(0.80–1.51)	(0.79–1.49)	(0.77–1.47)	(0.80–1.51)
Free sugars (g/day)	Mean	19.5	20.0	19.4	16.3	19.0
	Median	17.8	18.4	17.7	14.2	17.4
	(P25–P75)	(11.8–25.2)	(12.4–25.8)	(11.8–25.1)	(8.8–21.5)	(11.6–24.6)
Aflatoxins (ng/kg bw/day)	Mean	0.065	0.073	0.065	0.052	0.090
	Median	0.049	0.054	0.047	0.038	0.066
	(P25–P75)	(0.0–0.95)	(0.0–0.109)	(0.0–0.094)	(0.0–0.078)	(0.0–0.136)
*Bacillus cereus* (log CFU/day)		2.5	2.1	2.6	<0 *	3.3

P25: 25th percentile; P75: 75th percentile; bw: body weight; BC: breakfast cereals; IC: infant cereals; * <0 log ufc/g equals <1 ufc/g.

**Table 6 nutrients-13-03127-t006:** Number of prevented or extra cases of type 2 diabetes (DM2), cardiovascular diseases (CVD) and hepatocellular carcinoma (HCC) due to consumption of BC and IC by Portuguese children between 6 and 35 months of age in each alternative scenario.

	100% BC	100% IC	Best BC	Worst IC
DM2 (fiber)				
Number of cases/100,000/year * (95% CI)	−0.714	0.211	−6.618	1.272
	(−0.998; −0.482)	(0.143; 0.295)	(−9.229; −4.489)	(0.858; 1.782)
CVD (fiber)				
Number of cases/100,000/year * (95% CI)	−0.160	0.047	−1.469	0.288
	(−0.253; −0.072)	(0.021; 0.074)	(−2.288; −0.668)	(0.127; 0.458)
HCC (AFB_1_)				
Number of cases/100 k/year ** (95% CI)	0.00073	0.00065	0.00077	0.00090
	(0; 0.0027)	(0; 0.0023)	(0; 0.0028)	(0; 0.0033)
Number of Gastrointestinal disease (*Bacillus cereus*)	0	0	0	0

* Difference in incident cases between alternative and reference scenarios; BC: breakfast cereals; IC: infant cereals, ** Number of extra incident cases due to aflatoxin exposure in each scenario.

**Table 7 nutrients-13-03127-t007:** Overall DALY difference (ΔDALY) estimates for each alternative scenario, integrating risks and benefits associated with consumption of breakfast cereals (BC) or infant cereals (IC) by Portuguese children aged between 6 and 35 months.

	100% BC	100% IC	Best BC	Worst IC
Sum of ΔDALY (per 100 k individuals) (95% CI)	−0.486	0.143	−4.473	0.877
	(−0.727;−0.262)	(0.078;0.213)	(−6.614;−2.449)	(0.471;1.317)

BC: breakfast cereals; IC: infant cereals.

**Table 8 nutrients-13-03127-t008:** Prevalence of Portuguese children aged 6–35 months with an intake of sodium and free sugars, from all dietary sources, above the DRV in the reference and alternative scenarios (semi-quantitative approach).

DRV	Reference	100% BC	100% IC	Best BC	Worst IC
**Sodium (UL)**					
1500 mg/day	24.8%	25.7%	24.3%	23.4%	25.4%
**Free sugars (RI)**					
5% TEI	80.0%	81.9%	79.8%	69.7%	79.0%
10% TEI	29.4%	31.6%	29.2%	16.0%	27.7%

DRV: Daily Reference Value; UL: upper-limit; RI: recommended intake; TEI: total energy intake; BC: breakfast cereals; IC: infant cereals.

## Data Availability

Data available on request.

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
