# Peer review of "Risk-Benefit Assessment of Cereal-Based Foods Consumed by Portuguese Children Aged 6 to 36 Months—A Case Study under the RiskBenefit4EU Project"

_nutrients, 2021, doi:10.3390/nu13093127_

Round 1
Reviewer 1 Report
Title
Please rearrange title to underline children age from 6 to 36 months
Abstract (….. exclusive BC consumption …)
Please rearrange sentence, the word exclusive leads to wrong suggestions (eg how about fruits etc)
Introduction
Please provide key definition (eg according to CDC) about infant and toddlers age
Please provide more information to distinguish between IC and BC cereals (eg constituents, single grain cereals, time of introduction to children menu according to their age etc)
Please provide scientific opinions of public health organizations and pediatricians about the proposed so far children menu up to the age of 36 months (rg recommendations about the total energy intake etc)
Define current consumption (as mentioned in abstract)
Line 203
Please provide the meaning of LOD the first time it mentioned
Questions
How about food allergies from cereal consumptions ?
Why authors did not take into consideration the use of milk (consumed mainly with cereals) and its associated hazards in the semi-quantitative RBA ?
Author Response
Title
Please rearrange title to underline children age from 6 to 36 months
Thanks to the reviewer for this comment. We have adjusted the title to the following: “Risk-benefit assessment of cereal-based foods consumed by Portuguese children aged 6 to 36 months – a case study under the RiskBenefit4EU project”
Abstract
(….. exclusive BC consumption …)
Please rearrange sentence, the word exclusive leads to wrong suggestions (eg how about fruits etc)
Again thanks for this relevant comment. We have arranged the sentence accordingly to: “Portuguese children can benefit from exchanging intake of IC to BC, if the BC consumed have an adequate nutritional profile in terms of fiber, sodium and free sugars along with levels of aflatoxins reduced as much as possible.”
Introduction
Please provide key definition (eg according to CDC) about infant and toddlers age
We have in line 50-51 now highlighted the age ranges for infants and toddlers, respectively: “Cereal-based foods, including breakfast cereals (BC), i.e. ready-to-eat processed grains, and infant cereals (IC), i.e. processed cereal-based foods that are or have to be reconstituted with water, milk, or other appropriate liquids, are among the first solid foods introduced to infants (age 0-1 (www.cdc.gov)) and important components of the diet for toddlers (age 0-3 (www.cdc.gov)) [1,2].”
The age range is also highlighted in line 69-71: “In Portugal, previous studies have reported that ICs and BCs, constitute a considerable proportion of the diet of infants and toddlers (children under three years of age) [5,6,20–22].”
Please provide more information to distinguish between IC and BC cereals (eg constituents, single grain cereals, time of introduction to children menu according to their age etc)
Thanks for pointing out this. We have attempted to distinguish further between BC and IC in line 71-75): “ICs are traditionally considered suitable as the first solid foods introduced to infants 6 months of age. BCs are also marketed for infants but considered more suitable for toddlers of older age. According to products available on the Portuguese market, BC usually contain several types of grains and more fiber and less sodium and added sugar than ICs [5,6, 22].”
Also table 2 provide the nutritional composition of IC and BC, respectively, of selected products available on the Portuguese market.
Please provide scientific opinions of public health organizations and pediatricians about the proposed so far children menu up to the age of 36 months (rg recommendations about the total energy intake etc)
Thank you for the suggestion to include this relevant information. We have now included the energy intake requirement for up to 36 months of age in the introduction. Elsewhere in the text, the recommended age of introduction of solid foods is described. We agree that the total diet of toddler and infants are of relevance, however further specific information we believe is outside the scope of our study. Please note, that the background intake of fiber, sodium and sugar from other sources is included in the assessment. The following text has been included in line 79-84: “Likewise, the consumption of IC and BC by young children could represent a significant contribution to the daily intake of sodium and sugar, but also to the overall energy intake (recommended energy intake for children up to 36 months is according to Food and Agriculture Organization (FAO) up to 427 KJ/kg bodyweight per day [25]) , and act as risk factors for development of future non-communicable diseases.”
Define current consumption (as mentioned in abstract)
We have now included how we define the current consumption of IC and BC in line 171-173: “The current consumption is defined as the consumption of IC and BC reported in the latest survey of the Portuguese dietary habits [22], as described below under 2.3.)”
Line 203
Sorry, we don’t know what is referred to here?
Please provide the meaning of LOD the first time it mentioned
Thanks, this is now indicated in line 213.
Questions
How about food allergies from cereal consumptions ?
The development of food allergies is of course relevant when investigating infant/toddler diets. We have included in the discussion line 452-454 that food allergies were not considered, which is a limitation: “also food allergies are relevant health outcomes when investigating the health impact of infant and toddler foods but have been disregarded in this assessment.”
Why authors did not take into consideration the use of milk (consumed mainly with cereals) and its associated hazards in the semi-quantitative RBA ?
We have decided to just focus on the cereal-based foods. The use of milk could be diverse (i.e. different types of milk), and some of the infant cereals already include milk in the list of ingredients. So, in order to remain focused on the main objective, we have decided in just assess the risk and benefits associated to the cereal-based products.
Reviewer 2 Report
This is a case study discussing about whether Portuguese children can benefit from exclusive BC consumption. I rather named a cross-sectional study design for this research. This study used a population-based, nationally RiskBenefit4EU project. I think the topic is important and contributive to the public health with an empirical approach quite valuable for public policy professionals.
Major concerns:
- The research appropriately used beta distribution the prevalence based on the percentage of positive values. Based on my knowledge, dispersion test should be provided to approval evidence of selected method in this study. By the way, binomial model also had the similar results?
- Table 1 didn’t present characteristics of 5,811 participants. The authors should provide more general/additional characters of studied subjects.
- Please use power analysis to statement adequate sample size in this study.
- I am not familiar the formulas used in this study. The authors should provide more general/additional description of formulas, respectively.
- Lastly, the authors only briefly discuss limitations, acknowledging that the main limitation is the cross-sectional design. They should elaborate on how the use of this design is subject to Incidence-Prevalence bias, also known as Neyman bias, and how that might influence their findings.
- Some references are outdated (or error) and should be updated accordingly. For example, “Error! Reference source not found.” In lines 103, 174, 319 etc.
Author Response
Reviewer 2
This is a case study discussing about whether Portuguese children can benefit from exclusive BC consumption. I rather named a cross-sectional study design for this research. This study used a population-based, nationally RiskBenefit4EU project. I think the topic is important and contributive to the public health with an empirical approach quite valuable for public policy professionals.
Thank you for this view of the relevance of our study. We would like to clarify that our study is not a cross-sectional population-based study, but an impact modelling study combining population intake and exposure data from the Portuguese dietary survey with dose response relationships for the included exposure-health outcome pairs that is derived from the scientific literature. This means that we did not derive any estimates of risk ratios from statistical analyses of the population in the dietary survey.
Major concerns:
- The research appropriately used beta distribution the prevalence based on the percentage of positive values. Based on my knowledge, dispersion test should be provided to approval evidence of selected method in this study. By the way, binomial model also had the similar results?
We thank the reviewer for this question, but hope that the following provides the necessary clarifications. Beta distributions are commonly used probability distributions to describe uncertainty about the true value of a proportion, such as prevalence (Vose, 2000). It is well established that when used to express prevalence, the Beta distribution can be defined by the two parameters, alpha and beta (written as Beta(alpha, beta)), with alpha = x + 1 and beta = n - x + 1, where x is the number of positive events out of n trials. As n increases, the degree of uncertainty (the width of the distribution) about the estimated proportion (x/n) decreases.
- Table 1 didn’t present characteristics of 5,811 participants. The authors should provide more general/additional characters of studied subjects.
Thank you for the comment. Table 1 provides an overview of the selected components and their associated health outcomes that are accounted for in our study. In section 2.3. the data used in the model is provided, herein some of the characteristics of the National Food and Physical Activity Consumption Survey including the number and age range of the individuals in the survey and the method of dietary records. This survey is used to estimate the intake of IC and BC and associated exposure to the included components. We have added the number of individual in the survey aged 6-36 months (line 212: “In this study, food consumption data from the 779 children aged 6 to 36 months from the IAN-AF 2015-2016 sample was included.”), but have referred to publications that in higher detail describe the survey and its representativeness of the Portuguese population.
- Please use power analysis to statement adequate sample size in this study.
We don’t believe that power analysis is relevant to apply in the current study. The power analysis has been applied when the sample size/number of participants in the Portuguese dietary survey was determined, as presented in detail in the methodological publication of the IAN-AF 2015-2016 survey (please, see reference [21]). We use this dataset as this is the best available study to assess the daily intake of IC and BC in the Portuguese population.
- I am not familiar the formulas used in this study. The authors should provide more general/additional description of formulas, respectively.
Thanks for this comment. We have tried to add information about each calculation step in the main manuscript where we found it relevant. In addition, we have referred to the supplemental materials where the full models are represented for each domain (nutrition, tox and microbiology). Specifically we have added:
Line 234-248:
“In order to estimate the relative risk (RR) of the current and each alternative intake scenarios, we performed the following calculations. The distribution of fiber intake of children aged 6 to 35 months, which was derived from the National Food and Physical Activity Consumption Survey [22], was divided into quartiles, with each quartile representing a consumption class (1-4). The median of each class represented the intake of each class, respectively. RRs for DM2 and CVD derived from the literature (Table 3) was used to estimate a RR for each class, assuming a RR of 1 at zero exposure and a log-linear association between exposure and RR [56] (appendix B and Table S2). Thus, the log-linear slope, β, and the RR for each class, j ∈ {1,2,3,4}, in each scenario, i ∈ {1,2,3,4,5} (i.e. the reference and the four alternative scenarios), were calculated according to the following equations:
, (2)
, (3),
where RRliterature are the RR for DM2 and CVD per Dose reported in the literature (Table 3), and RRi and exposurei are the RR and intake of fiber in each scenario.”
Line 249-257:
“The Potential Impact Fraction (PIF) is a measure of the of the proportional change in disease burden after a change in exposure of a related risk factor, in this case change in exposure to fiber from the current to alternative scenarios, and was calculated for each alternative scenario [57]. We applied the RR shift methodology [57], which assumes that the change in exposure is described by a change in the RR of the categories, while keeping the proportion in each category constant:
, (4)
where RRref is the RR in the reference scenario and RRalt is the RR in each alternative scenario.”
- Lastly, the authors only briefly discuss limitations, acknowledging that the main limitation is the cross-sectional design. They should elaborate on how the use of this design is subject to Incidence-Prevalence bias, also known as Neyman bias, and how that might influence their findings.
Thank you for this comment. As stated in our first comment to the reviewer, the present study is a applying a risk assessment approach combining intake/exposure estimates from a cross-sectional study (the Portuguese dietary survey) and combine this with dose response relationships for each exposure-health outcome pair derived from the scientific literature. Therefore, we do not believe that Neyman bias is relevant to consider in our study, but rather in the underlying population-based studies retrieved from the literature. We do propagate uncertainty from various model parameters applying distributions and Monte Carlo simulation techniques.
- Some references are outdated (or error) and should be updated accordingly. For example, “Error! Reference source not found.” In lines 103, 174, 319 etc.
We are sorry about this. However, in the version available in the submission platform, no error messages are present. Perhaps, it is just present in the pdf version of the manuscript sent to the Reviewers?
Round 2
Reviewer 2 Report
I find the paper has improved in the revised manuscript. Finally, I evaluate your response and examine the consistency among reply and revision on texts of revised manuscript.
Author's Notes: Thanks for revision. Please determined the type of study design and add it to method section.
Major concerns
Response 1: Thanks for revision. No further comments.
Response 2: Thanks for revision. No further comments.
Response 3: Please add your response to appropriate location of method section.
Response 4: The authors’ response really caused confusion to me. “We would like to clarify that our study is not a cross-sectional population-based study, but an impact modelling study…” contrasted to “the present study is a applying a risk assessment approach combining intake/exposure estimates from a cross-sectional study (the Portuguese dietary survey)…..”. I appeal to the authors to clarify this contradicted response.
Response 5: Noticed.
Author Response
I find the paper has improved in the revised manuscript. Finally, I evaluate your response and examine the consistency among reply and revision on texts of revised manuscript.
Author's Notes: Thanks for revision. Please determined the type of study design and add it to method section.
Major concerns
Response 1: Thanks for revision. No further comments.
Thank you.
Response 2: Thanks for revision. No further comments.
Thank you.
Response 3: Please add your response to appropriate location of method section.
Thank you for the suggestion. This information was added in lines 194-196, as:
Sample size of participants was determined, considering the power analysis, and as presented in detail in the methodological publication of the IAN-AF 2015-2016 survey [21].
Response 4: The authors’ response really caused confusion to me. “We would like to clarify that our study is not a cross-sectional population-based study, but an impact modelling study…” contrasted to “the present study is a applying a risk assessment approach combining intake/exposure estimates from a cross-sectional study (the Portuguese dietary survey)…..”. I appeal to the authors to clarify this contradicted response.
Thank you for the opportunity to clarify this aspect. In the present study, we used already obtained and reported consumption data from a cross-sectional study (IAN-AF). We did not perform the cross-sectional study under the scope of this study, neither this study is a cross-sectional study.
Response 5: Noticed.
Thank you.